# A Narrative Review of Telemedicine in Latin America during the COVID-19 Pandemic

**DOI:** 10.3390/healthcare10081361

**Published:** 2022-07-22

**Authors:** Genesis Camacho-Leon, Marco Faytong-Haro, Keila Carrera, Marlyn Molero, Franhe Melean, Yuliana Reyes, Hans Mautong, Ivonne De La Hoz, Ivan Cherrez-Ojeda

**Affiliations:** 1División de Estudios para Graduados, Facultad de Medicina, Universidad del Zulia, Maracaibo 4001, Venezuela; genesisc23@gmail.com (G.C.-L.); ivodelahoz11@gmail.com (I.D.L.H.); 2Division of Clinical and Translational Research, Larkin Community Hospital, South Miami, FL 33143, USA; 3Sociology and Demography Department, The Pennsylvania State University, University Park, PA 16802, USA; 4Ecuadorian Development Research Lab, Daule, Guayas 090656, Ecuador; 5School of Health, Universidad de Especialidades Espíritu Santo, Samborondón, Guayas 0901952, Ecuador; hmautong@gmail.com; 6Gastroenterology, Universidad de Oriente, El Tigre, Anzoátegui 6050, Venezuela; keilacarrera@gmail.com; 7Médicos Unidos Por Venezuela, Caracas, Distrito Capital 1000, Venezuela; marlynmolero@gmail.com (M.M.); franhe_m@yahoo.com (F.M.); yuli939@gmail.com (Y.R.); 8Respiralab Research Group, Guayaquil, Guayas 090512, Ecuador

**Keywords:** telemedicine, COVID-19, Latin America, access to health, right to health

## Abstract

Background: The COVID-19 pandemic greatly impacted traditional health structures, posing new challenges in an unprecedented health crisis. Telemedicine services were implemented in countries with robust digital platforms to reduce hospital attendance while continuing to provide medical care. This study aims to determine how telemedicine services have been used as a tool to ensure the right to health in Latin America during the pandemic. Materials and methods: We conducted a narrative review in which words such as telemedicine, COVID-19, Latin America, access, and right to health, were searched on scientific medical datasets such as PubMed and SciELO. Additionally, we reviewed legislation in the Latin American health domain regarding the administration and transmission of digital data. Results and conclusions: Several countries have used telemedicine to reduce the saturation of healthcare systems and increase patient access. Issues such as broadband access for low-income populations and adequate legal regulations for transmitting and storing confidential data must be addressed to improve telemedicine use in Latin America.

## 1. Introduction

The healthcare systems in Latin America are fragmented between public and private institutions and demonstrate variations in regional numbers and access to healthcare facilities and medical personnel. To ensure healthcare coverage across the region, more resources and effort are needed to cover the Latin American populations’ needs.

Beginning in November 2019, the rapid transmission of COVID-19 globally placed further burden on healthcare systems and constraints on populations’ access to appropriate medical care. Under these circumstances, telemedicine has emerged as a tool to reach out to more people and relieve hospitals overcrowded with COVID-19 cases [1,2,3].

Telemedicine has become more prevalent in Latin American countries, broadening access to care in both public and private systems, allowing patients to access care at a lower cost than an “in-person” visit to a medical office, and lowering the risk of contracting COVID-19 while in a hospital setting [2,4,5,6]. It has even been found to aid in early diagnosis of diseases and reduce recovery time [2]. Within Latin America, Ecuador, Chile, and Uruguay have been pioneers in the application of telemedicine services, including the development of legal frameworks to regulate its use.

For this article, we conducted a narrative review by searching scientific medical websites such as PubMed and SciELO for topics such as telemedicine, COVID-19, Latin America, health access, and the right to health. Only two keywords had to be present in the source to be considered for inclusion. The only condition for exclusion was that the data came from outside the United States. In addition, we reviewed Latin American healthcare regulations related to data management and transmission.

This review aims to summarize and discuss the important contributions and constraints of telemedicine services in the Latin American region during the COVID-19 pandemic and suggests pathways forward to effectively expand telemedicine access across the region [2,6].

## 2. Demographic Characteristics of the Population in Latin America

Since the 1960s, there has been a shift in Latin America’s demographic dynamics. According to ECLAC, the fertility rate decreased by 62.8 percent between 2015 and 2020 compared to the 1970s’ figures. By 2018, the regional population was estimated to be around 652 million, with children and adolescents under 15 accounting for 25% of the population. Today, average life expectancy in Latin America is 76 years [6,7].

It is estimated that up to two-thirds of the region’s population resides in cities. The migration of people from the countryside to cities poses challenges in addressing public policies, including healthcare. Rural areas in Latin America are at a social and economic disadvantage, with limited essential services such as electricity and health systems. Latin American countries also face the additional challenge of sheltering and serving refugees from neighboring or nearby countries, further saturating health systems. To support the high demand for healthcare services across urban and rural areas of Latin American countries, there is a need to implement new strategies, such as information and communication technologies (ICT) [8,9,10].

Latin America’s demographic composition can be viewed as advantageous for the use of new technologies in the implementation of healthcare [11,12,13], as the presence of many young people in the region may reflect a greater willingness to use these technologies. However, older adults and residents of rural areas may require educational processes to adapt to new technological tools such as telemedicine services [6,14].

## 3. Health Care Quality in Latin America

The right to health is one of the fundamental principles enshrined in the Universal Declaration of Human Rights [1]. The World Health Organization has determined that health “is a state of complete physical, mental and social well-being and not merely the absence of disease or infirmity. The enjoyment of the highest attainable standard of health is one of the fundamental rights of every human being without distinction of race, religion, political belief, economic or social condition” [2].

By this definition, “health” is an overarching state of well-being, not only limited to medical care access in health care centers. This paper will focus on physical health as well as the regulations that help to ensure health coverage across Latin America [3]. One of the United Nations’ Sustainable Development Goals is universal health coverage, which includes providing high-quality, standardized health care to all people [15]. Healthcare quality determines whether the services provided are acceptable and adequate for the population’s needs [16,17]. The heterogeneity of Latin American healthcare systems makes it difficult to evaluate healthcare quality in a generalized manner; however, relevant and common factors among the nations of this region can be mentioned [18].

Latin American healthcare systems are fragmented between public and private institutions. In Latin America, the quality of healthcare systems is considered superior in private institutions as it has shorter wait times to access medical care and typically has better infrastructure than public institutions [18]. By way of example, a study conducted by Aravena and Inostroza from 2011 to 2012 reported that 67.50 percent of Chileans rated their country’s health system as poor where 68.20 percent of total respondents were covered by FONASA, the country’s public health system [19].

Investment in Latin America’s public healthcare systems is estimated to be low, resulting in even greater inequity in care across public and private institutions, which has become more apparent due to the effects of the COVID-19 pandemic [20,21]. Cuba has been identified as the country that invests most in its healthcare system, with up to 13.3 percent of its gross domestic product (GDP) invested in healthcare policies, while Venezuela invests the least, with 1.1 percent of its GDP invested in the matter. [22,23]. Health expenditure, which represents the payment per family for health care, is inversely proportional to government investment in health. In countries such as Venezuela, it amounts to 62.99 percent of family annual income, while in Guatemala, it is 54.13 percent, in stark contrast to Colombia, which spends 16.31 percent, and Argentina, which spends 15.02 percent. Access to healthcare is further impacted by the infrastructure and human capital available to support the region’s populations. Brazil is the country that has the most healthcare facilities in the region, with an estimated 7400 hospitals in 2020, while countries such as Venezuela, Honduras, and Nicaragua have fewer healthcare facilities available. Cuba had the highest number of doctors, 8.4 per 1000 inhabitants in 2018. By comparison, In 2018, Brazil had a total of 2.16 doctors per 1000 inhabitants, while Honduras and Guatemala had 0.5 doctors or less per 1000 inhabitants in 2020, making them the countries with the fewest doctors in the region [23].

A fragmented healthcare system and varying rates of healthcare facilities and medical personnel make achieving healthcare for all in Latin America difficult. These shortcomings are reflected in the massive toll that the Latin American region paid with the COVID-19 pandemic. Latin America had 1.7 million deaths as of 3 May 2022 (over 27 percent of deaths worldwide). Deaths were highest in Brazil, Mexico, Peru, Colombia, and Argentina. To address unequal healthcare access, telemedicine offers a way to supplement coverage.

## 4. Telemedicine Use in the Region

The World Health Organization (WHO) defines telemedicine as “the delivery of health care services, where distance is a critical factor, by all health care professionals using information and communication technologies (ICT) for the exchange of valid information for the diagnosis, treatment and prevention of disease and injuries, research and evaluation, and for the continuing education of health care providers, all in the interests of advancing the health of individuals and their communities” [24].

After analyzing data from the Latin American region, the WHO and Pan American Health Organization concluded, in 2017, that eHealth standards had not been appropriately represented through scientific study. Publications and analyses are limited to pilot tests, contraining the development of strategies from ensuring the implementation of information and communication technologies in these countries.

In February 2021, Saigi-Rubio et al. published a study in which they synthesized public policies and the various factors influencing the region’s development of international telemedicine services. They determined that ICT could close the health gap, reduce the inequity between countries and their populations, and ensure compliance with the United Nations’ development goals; however, despite the numerous benefits that could be reported, there are still barriers to its implementation. This study also identified as a significant issue the lack of clear telemedicine guidelines and the lack of an accessible technological framework. Additionally, they found limiting legislative frameworks in most countries, even with advances due to the pandemic’s health demands [25].

The advancement of telemedicine has many advantages, but it also has some drawbacks related to advanced technologies, connectivity quality, digital inclusion, and legal and ethical aspects of healthcare [26]. Table 1 shows the current state of telemedicine-related laws in select Latin American countries, which inequalities they solve, and examples of telemedicine usage.

### 4.1. Argentina

Telemedicine has been gaining traction in Argentina for over a decade. On 15 October 2020, a bill was passed to regulate the use of telemedicine, including rules for easy access to this tool. They regulated telemedicine costs and Internet access. Before that, a legislative decision dating back to 2012 dictated the acceptance of the transfer of health data to international institutions using telemedicine [52]. In 2020, telemedicine services began to address the health crisis, providing remote assistance and achieving a total of 83,000 teleconsultations as of October of that year. Since then, telemedicine services have continued to be used successfully [30].

### 4.2. Bolivia

Although the telehealth program has not yet been fully regulated, the government has used teleconsultations since 2013. Since the pandemic, telemedicine services have been implemented in the public sector to educate, evaluate, and analyze COVID-19 cases [53]. On 13 February 2020, the Bolivian public health care system launched telemedicine services to answer questions about SARS-CoV-2, detect suspected cases, and follow up on cases. They completed approximately 200,000 teleconsultations in the first 100 days of the plan’s implementation [31,32]. According to official statistics for January 2022, the population was still using this service, with approximately 1400 calls handled daily [54].

### 4.3. Brazil

Since 2011, the use of ICT has expanded rapidly with telehealth online programs, which a group of cardiologists initially implemented to provide services to underserved areas. In recent years, e-medicine services have expanded to all specialties. However, the nation has focused on digital health within the country and, according to Article 199 of its constitution, prohibits foreign healthcare companies and facilities from operating within the borders of Brazil, which may limit healthcare access across the country. During the COVID-19 pandemic, telemedicine services as a public health strategy have assisted in making better use of resources and increasing hospital bed availability [25,33].

### 4.4. Chile

In the context of the pandemic, an exempt Resolution N. 277/2011 was modified, establishing that remote access to healthcare is authorized as long as records of each telemedicine appointment are maintained in patient files. As a result, the use of ICTs is widespread in Chile; consultations between healthcare providers and patients are performed synchronously [35]. In September 2021, it was proposed to promote a project to extend telemedicine as an easy-to-access tool even after the pandemic, following a 900 percent increase in its use after the pandemic’s start [36].

### 4.5. Colombia

Telemedicine has been used in Colombia since 2010 when Law 1419 for telehealth was enacted. The Ministry of Health and Social Protection established more explicit parameters for telemedicine in 2019. During the pandemic, this tool expanded in the country, raising virtual appointments to 100 million by May 2021, reducing the overload in health centers, and improving care to citizens [39,55,56].

### 4.6. Ecuador

In 2007, Ecuador established FUNDETEL, an organization that regulates eHealth policies. While there is still a lack of a comprehensive regulatory framework, telemedicine services are being used in both the public and private sectors. In March 2020, President Lenin Moreno, launched the SaludEC app to address the healthcare crisis caused by the SARS-CoV-2 pandemic. Through this app, the Government Health Ministry managed the health of patients with various pathologies, including COVID-19 [40].

### 4.7. Honduras

The government is developing a plan to expand the use of telemedicine with support from the International Development Bank. The legal frameworks for this plan will be the constitution, the law on the use of personal data, and the law on electronic commerce. Additionally, they launched a pilot plan to provide triage services in October 2021 for pandemic and chronic disease management. However, no data on the national implementation of these services has been reported yet [41].

### 4.8. Mexico

Telehealth services have been widely used as a public policy promoted by the national government since 2015 in its public health care system [43]. These services ensured that many citizens had access to health care during the pandemic without the need to implement new methodologies along the way, focusing instead on promoting tools that already existed [44]. The widespread use of telemedicine services during the pandemic prompted the presentation to the Senate of an initiative for the regulation of remote medical care, proposing a reform of the General Health Law to continue sustainably using these services to meet the future challenges of demographic growth and need in the health sector [57].

### 4.9. Peru

The country has made great strides in the area of telemedicine. Since 2016, various educational and non-governmental organizations have collaborated and participated in the NAPO project, which provided broadband and telemedicine tools to indigenous populations in the Amazon jungle, serving over 3000 inhabitants of eight isolated communities and facilitating their access to health care. Furthermore, they provided telemedicine services to 13 rural health centers serving over 8500 people, and several campaigns were promoted in 2020 to encourage the use of telehealth amid the health crisis [47]. Ministerial Resolution N° 081-2022-MINSA was established in February 2022, which specified the technical specifications of the essential equipment for telemedicine services. One of the resolution goals is to connect at least 50% of health facilities so they can provide telemedicine services [46].

### 4.10. Uruguay

Uruguay enacted a telemedicine law in April 2020, which, when combined with the nationwide use of fiber optics for easy Internet access, ensured access to healthcare and allowed up to 86 percent of COVID-19 positive cases to be attended to at home remotely. In addition, these services provide care for chronic diseases while reducing the overload on the healthcare system [48].

### 4.11. Venezuela

President Nicolás Maduro proposed creating a national and international telemedicine plan on 8 October 2021, which has not yet been published in the national registry. The public system has not yet presented a legislative and infrastructure platform to promote the use of these technologies [58]. On the other hand, several regional professional associations have established community support networks for telehealth since the pandemic. One example is “LLAMADAS SOS,” a group formed at the Universidad Central de Venezuela’s medical school to provide remote care to patients suffering from respiratory symptoms [51]. These practices have been provided in the absence of a clear legal framework, as this country lacks direct regulations for using information and communication technologies [59].

### 4.12. All Other Latin American Countries

Most of the remaining Latin American countries have rules that regulate health and digital markets, but without clearly establishing regulations and limitations on telemedicine practices and without providing Internet access facilities to their population, which creates a barrier to health care [25].

## 5. The Benefits and Limitations of Using Telemedicine Services in Latin America

The increasing use of telemedicine in Latin America has reduced inequities in access to healthcare by promoting care in isolated or underserved communities. Rural areas with mobility and communication issues have seen particular benefit by reducing the neglect of indigenous and rural people’s healthcare needs and avoiding overcrowding of health centers during the pandemic [24,25,47].

Since the declaration of the COVID-19 pandemic, most countries in the region have experienced interruptions in providing health services for non-communicable diseases. The use of telemedicine has facilitated the management of mild and moderate cases from home in countries such as Uruguay, Mexico, and Ecuador. In addition, emergency hotspots were activated to remotely assess the condition of patients affected by COVID-19 and other diseases, avoiding exposure of patients to the virus in health centers and reducing the number of in-person visits for physicians [40,44,48].

However, although many countries have simplified access to health care, some limitations remain, including the limitations inherent to this service, such as the lack of physical examination which can allow a more thorough diagnosis [24].

There are also other limitations, such as the high cost of remote monitoring equipment, making access difficult in the region. For example, the minimum wage in Chile, one of the most economically stable countries, is around USD 410/month, and the minimum wage in Venezuela, the country with the highest inflation, was calculated at USD 1/month, marking extreme economic limitations for the acquisition of specialized equipment for use at home. The economic issue is also significant because many families do not have access to smartphones, and the cost of data plans for these phones is prohibitively expensive [60,61,62].

Similarly, while the Latin American region’s average literacy rate is estimated to be around 94 percent, which facilitates the use of new technologies, Internet access, however, is limited. By 2021, less than half of the households in all Latin America had fixed broadband services [63,64]. In Brazil, two-thirds of the population has access to the Internet, which is associated with increased use of mobile telephones and social networks, but this does not always imply broadband access [65,66]. Another significant issue is the instability of the electricity system in some countries, such as Venezuela, where two national blackouts occurred in 2019, leaving more than 20 million people without power, and where the electricity system has declined significantly over time [67].

Additionally, a lack of laws regulating telemedicine privacy policies puts into question patient confidentiality. One cross-sectional study from Cherrez-Ojeda et al. in Ecuador found that the main concern of most professionals participating in the study (83.6%) is regarding privacy policies and confidentiality [68]. Even though FUNDETEL has overseen eHealth policies since 2007, these concerns have persisted, highlighting the need to improve the legal regulation of ICT practices and the education of patients and health care providers about the norms for its use [40]. This concern should be explored in future studies in countries across the region.

## 6. Conclusions

The COVID-19 pandemic has forced regional governments to confront inequities in access to healthcare in their respective countries. Telemedicine services have emerged as a useful tool for delivering care in and across the region. Telehealth services allow medical practitioners to reach out remotely to larger swaths of the population, in both urban and rural areas, as well as shorten the time it takes for patients in overcrowded hospitals to receive medical attention. Compared with other Latin American countries, Chile, Ecuador, and Uruguay are at the forefront of using these services, laying the groundwork for the rest of the region’s use of telemedicine services.

However, despite the benefits of telemedicine, many countries in the region lack the legislation to appropriately regulate this relatively new form of digital healthcare and little research exists on strategies to deploy telemedicine across the diverse areas of the region. Additionally, physical and logistical barriers still exist, such as power outages, a lack of broadband Internet access, and low smartphone access in some populations. Finally, there are no clear guidelines regarding the recommendations for the use of electronic devices in telemedicine in most Latin American countries. This knowledge gap may warrant the need to explore different electronic tools and their efficacy to deliver electronic consultations. Thus, providing a better orientation of what electronic tools may be more useful and suitable for the region.

The COVID-19 pandemic quickly stimulated the use and expansion of telemedicine services in Latin America; however, further research is needed in order to determine the current effectiveness of the implementation of telemedicine in the region and recommend strategies for its continued expansion and regulation.

## Figures and Tables

**Table 1 healthcare-10-01361-t001:** Telemedicine laws, usage, and inequalities solved due to telemedicine in Latin American countries.

Country	Related Laws	Inequalities Resolved	Examples of Telemedicine Usage during the Pandemic
Argentina	- Telemedicine Regulation Bill, passed in 2020 [27]. - Law No. 27553 for electronic or digital prescriptions [28].	The number of public hospitals offering telemedicine services was doubled, resulting in greater population coverage. Furthermore, provincial health centers were outfitted with technological equipment to provide this service [29].	By October 2020, 83,000 teleconsultations were conducted [30].
Bolivia	- There is currently no regulatory framework in place for the use of telemedicine services.	The introduction of telemedicine reduced saturation in public health centers, facilitating population access [31].	From February to May 2020, 200.000 teleconsultations were conducted [32].
Brazil	- Federal Council of Medicine Law 1643/2002 defines telemedicine [29]. - Law 13.989 governs telemedicine practice for the duration of a health emergency [29].	Due to the use of telemedicine services there was an increase in hospital bed availability and an increase in services to underserved areas [33].	Through a mobile application, the municipality of Cotia accounted for more than 50,000 medical assistances in 2020 [34].
Chile	- Resolution N. 277/2011 authorizes remote access to healthcare while stating that records of the provision of healthcare services are required [35]. - A bill has been passed to extend the use of telemedicine services following the pandemic [36].	Prior to the pandemic, 65% of hospitals were using telemedicine services, which helped to keep treating COVID-19 patients without completely displacing attention from patients with chronic diseases [37].	According to the analysis of the Unit for the Generation of Statistics and Data of the Superintendency of Health, in the period between March and October 2020, 198,854 telemedicine consultations were carried out [29].
Colombia	- Law 1419 for telehealth, 2010 [38], which regulates the use of these services.	The overload in health centers was reduced, and the care to all citizens has been improved with telemedicine [39].	By May 2021, 100 million people had received health care via telemedicine services [39].
Ecuador	- There is no clear legal framework, but the FUNDETEL organization regulates eHealth cases. The government also generated the document “Consensus of recommendations for palliative care in the SARS-CoV-2/COVID-19 pandemic”, which authorizes the use of telemedicine [29].	Patients with COVID-19 as well as other diseases were treated with telemedicine services. Telemedicine services are being used in both the public and private sectors [29].	12,000 Ecuadorians received telemedicine care during the first six hours of the national plan to use telemedicine as a platform to combat the pandemic [40].
Honduras	Although there is no specific legal framework for telemedicine services, the constitution, the law on the use of personal data, and the law on electronic commerce are used on this matter [29].	In October 2021, public triage services for COVID-19 pandemic and chronic disease management were launched, intending to reduce waiting times for patients suffering from a variety of diseases [41].	During the COVID-19 pandemic quarantine from 12–30 March 2020, 54 tele-consultations were made to patients with respiratory pathology [42].
Mexico	There are no specific regulations for telemedicine; however, it has been included in public policies since 2015. In June 2020, a document titled “Unidad de contacto para interconsulta a distancia (UCID) México: atención a enfermedades crónicas” (Contact unit for remote interconsultation (UCID) Mexico: attention to chronic diseases) was published, which promotes the use of telemedicine in the treatment of chronic diseases [29].	For years, telemedicine services have been used to address various pathologies and demographic challenges in health, with the public sector managing to provide this service to the entire population [43].	More than 5.5 million COVID-19 telemedicine consultations were reported in 2020 alone [44].
Peru	- Ministerial Resolution 1010-2020-MINSA approves the Technical Document of the National Telehealth Plan [45]. - Ministerial Resolution N° 081-2022-MINSA, 2022,specifies the technical requirements of the equipment needed for telemedicine services [46].	These aided in increasing access to healthcare for remote communities, and during the pandemic health emergency. There was a special emphasis on remote medical guidance, telemonitoring, and mental health [29].	In 2016, a special project served 3000 residents of eight isolated communities by providing access to telemedicine services to 13 rural health centers [47].
Uruguay	- In April 2020, Law 19.869 was promulgated on the “Approval of the general guidelines for the implementation and development of telemedicine as a health service provision” [29].	This increased access to the healthcare system due to the wide use of telemedicine services by the population and high availability of doctors through these services [48].	By April 2020, up to 86% of COVID-19 positive cases were being treated remotely at home [48].
Venezuela	- The 2001 Computer Crimes Act governs the protection of digital information [49]—The TeleHealth law, enacted in 2015, regulates telemedicine services without establishing a clear infrastructure for them [50].	Patients with respiratory symptoms who were unable to visit a clinical facility received free remote care [51].	Volunteer physicians created a non-governmental hotline to treat patients with respiratory symptoms [51].

## Data Availability

Not applicable.

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
