# Peer review of "A Narrative Review of Telemedicine in Latin America during the COVID-19 Pandemic"

_healthcare, 2022, doi:10.3390/healthcare10081361_

Round 1

Reviewer 1 Report

The article is relly important, interesting and valuable. It describes problems of telemednicine in the South America. The article is written in logical menner. There is lack of wrong reasoning.The article is presented in an aesthetic manner. The authors have edited the article properly.The correct way of thinking was adopted. The conclusions are correct. The article is important for WHO.

Minor:

The serious lack of the article is the deficiency of drawings, formulas, and statistical formulations.

The article does not provide indications for the use of electronic devices for telemedicine. In subsequent works, the authors can select electronic tools for telemedicine consultations.

Author Response

Response to Reviewer 1 Comments

We are very grateful to Reviewer #1 for providing comments on the manuscript. We believe that the manuscript has improved significantly because of all the changes in the revision.

Point 1: The article is relly important, interesting and valuable. It describes problems of telemednicine in the South America. The article is written in logical menner. There is lack of wrong reasoning.The article is presented in an aesthetic manner. The authors have edited the article properly.The correct way of thinking was adopted. The conclusions are correct. The article is important for WHO.

Response 1: We thank Reviewer #1 for this comment. We truly believe will article will be a good reference for elaborating further telemdicine guidelines. 

Point 2: Minor: The serious lack of the article is the deficiency of drawings, formulas, and statistical formulations.

Response 2: We appreciate your comments. For a better understanding, a summary of the information reviewed has been added. Because this is a narrative review, no statistical formulas were included.

Point 3: The article does not provide indications for the use of electronic devices for telemedicine. In subsequent works, the authors can select electronic tools for telemedicine consultations.

Response 3: Thank you for your suggestion, we added more clarification about the lack of guidelines for indications and recommendations for the use of electronic devices in Telemedicine, we also suggest that further research is needed in order to explore different electronic tools and their efficacy for electronic consultations in the region. We wrote this in the second paragrpah of the “Conclusion”:

Additionally, there are no clear guidelines regarding recommendations for the use of electronic devices in telemedicine in most Latin American countries. This knowledge gap may warrant the need to explore different electronic tools and their efficacy to deliver electronic consultations. Thus, providing a better orientation of what electronic tools may be more useful and suitable for the region.

Variable ratings.

Response: We have noted that you marked with 4/5 in the statement if work is well organized and comprehensively described and if English is correct and readable. To improve readability, we hired  a native English speaker academic copy editor. They extensively worked on the manuscript to improve its flow, readabilidy, and clearness.

You also rated with 3/5 the statement of the work being scientifically sound and not misleading. We have also double-check well that the conclusion matched the narrative review in a better way.

Finally, you rated with 2/5 the statement about appopiate and adequate references to related and previous work. In resstruccuring the article, we have added more relevant references to previous work on Telemedicine in Latin America. However, we expect this paper to be one starting point for other researchers who want to explore empirically telemedicine in Latin America. Information collected about telemdicine in Latin America thorughtout this work is mostly available only for spanish or Portuguese speakers.

Reviewer 2 Report

This review aims to discuss the significance of telemedicine services in the Latin American area during the last SARS-CoV-2 pandemic. It also tries to consider the region's health access constraints and their impact on Latin American countries. At first hand, Latin America's demographic composition can be considered advantageous for the use of new technologies in the implementation of healthcare, as there are many young people in the region. Nonetheless, older citizens may have problems for accessing new technological tools and telemedicine services. Adding to that, investment in Latin America public health system is low, resulting in greater inequity in care. Then telemedicine has some advantages, but also some drawbacks related to advanced technologies, connectivity quality, digital inclusion, legal and ethical aspects, as well as economic problems and poverty that block the access to smartphones or other technical devices. Even it can be the case of health professionals using information technologies to improve communication with the health team and professional colleagues, but not with patients. Internet access can be limited, not always with broadband services. It's right to conclude that these items should be addressed to develop effective policies to improve access to health care. It is not right to conclude that the telemedicine services are a cost-effective way to provide care to the population. That has to be proved. And the current review only provides a brief summary of the telemedicine services in Latin America. To state that telemedicine services ara cost-effective, the review should provide an analytical study with figures and numbers. And that's not the case. 

Author Response

Response to Reviewer 2 Comments

We are very grateful to Reviewer #2 for providing comments on the manuscript. We believe that the manuscript has improved significantly because of all the changes in the revision.

Main Point (Background): This review aims to discuss the significance of telemedicine services in the Latin American area during the last SARS-CoV-2 pandemic. It also tries to consider the region's health access constraints and their impact on Latin American countries. At first hand, Latin America's demographic composition can be considered advantageous for the use of new technologies in the implementation of healthcare, as there are many young people in the region. Nonetheless, older citizens may have problems for accessing new technological tools and telemedicine services. Adding to that, investment in Latin America public health system is low, resulting in greater inequity in care. Then telemedicine has some advantages, but also some drawbacks related to advanced technologies, connectivity quality, digital inclusion, legal and ethical aspects, as well as economic problems and poverty that block the access to smartphones or other technical devices. Even it can be the case of health professionals using information technologies to improve communication with the health team and professional colleagues, but not with patients. Internet access can be limited, not always with broadband services. It's right to conclude that these items should be addressed to develop effective policies to improve access to health care. It is not right to conclude that the telemedicine services are a cost-effective way to provide care to the population. That has to be proved. And the current review only provides a brief summary of the telemedicine services in Latin America. To state that telemedicine services ara cost-effective, the review should provide an analytical study with figures and numbers. And that's not the case. 

Main Point (Prompt): It is not right to conclude that the telemedicine services are a cost-effective way to provide care to the population. That has to be proved.

Response 1: Thank you for your feedback, we appreciate your recommendation. We followed your suggestion and agree that a statement concluding that the implementation of telemedicine in Latin America is cost-effective is not accurate considering there is a lack of studies measuring this outcome. Therefore, we rephrased our conclusion and clarified the need of further research providing evidence to determine whether or not this may be cost-effective. Now our conclusion reads like this:

The COVID-19 pandemic has forced regional governments to confront inequities in access to healthcare in their respective countries. Telemedicine services have emerged as a useful tool for delivering care in and across the region. Telehealth services allow medical practitioners to reach out remotely to larger swaths of the population, in both urban and rural areas, as well as shorten the time it takes for patients in overcrowded hospitals to receive medical attention. Compared with other Latin American countries, Chile, Ecuador, and Uruguay are at the forefront of using these services, laying the groundwork for the rest of the region's use of telemedicine services.

However, despite the benefits of telemedicine, many countries in the region lack the legislation to appropriately regulate this relatively new form of digital healthcare and little research exists on strategies to deploy telemedicine across the diverse areas of the region. Additionally, physical and logistical barriers still exist such as power outages, a lack of broadband Internet access, and low smartphone access in some populations. Additionally, there are no clear guidelines regarding recommendations for the use of electronic devices in telemedicine in most Latin American countries. This knowledge gap may warrant the need to explore different electronic tools and their efficacy to deliver electronic consultations. Thus, providing a better orientation of what electronic tools may be more useful and suitable for the region.

The COVID-19 pandemic quickly stimulated the use and expansion of telemedicine services in Latin America; however, further research is needed in order to determine the current effectiveness of the implementation of telemedicine in the region and recommend strategies for its continued expansion and regulation.

Variable ratings.

Response: We have noted that you marked with 2/5 in the statement about the work being a significant contribution to the field. We expect this paper to be one starting point for other researchers who want to explore empirically telemedicine in Latin America. Information collected about telemdicine in Latin America thorughtout this work is mostly available only for spanish or Portuguese speakers.

We also noted that you marked with 2/5 in the statement about the work being well organized and comprehensively described. Also, you rated the paper English and our paper references appropiateness with 4/5. To improve readability, we hired a native English speaker academic copy editor. They extensively worked on the manuscript to improve its flow, readability, references relevances. and clearness.

We realized that you marked with 1/5 in the statement about the work scientifically being sound and not misleading. We suspect this could be due to comment about the misleading conlusion. We have addresed this point by changing the paper conclusion .

Reviewer 3 Report

A narrative review of telemedicine is in fashion under my experiences. Because of a horrible COVID-19 attack, such papers are written from many countries. 

As concrete comments, this paper has a good logical structure. But a level of completion is comparatively under medium. I propose specific comments.

Specific comments:

1. Impacts on telemedicine from COVID-19  are expressed too simply as case increases of telemedicine. In this aspect, this paper looks too plane. 

2. Additionally, for a better manuscript, you can propose aspects in which inequalities is resolved by telemedicine .    

3. You can provide damage report from COVID-19 in this region. From this damage, you must analyze needs of telemedicine compared with other regions. 

4. Telemedicine-related laws over Latin America  can be compared in a table for more readability. 

5. In line 64, the reference style is different compared with other styles.

6. Concrete examples of telemedicine usage can help the understanding of Latin America's status for other region's readers. In a current manuscript, writing style is somewhat strict.   

Author Response

Response to Reviewer 3 Comments

We are very grateful to Reviewer #3 for providing comments on the manuscript. We believe that the manuscript has improved significantly because of all the changes in the revision.

Background: A narrative review of telemedicine is in fashion under my experiences. Because of a horrible COVID-19 attack, such papers are written from many countries. As concrete comments, this paper has a good logical structure. But a level of completion is comparatively under medium. I propose specific comments.

Point 1: Impacts on telemedicine from COVID-19  are expressed too simply as case increases of telemedicine. In this aspect, this paper looks too plane. 

Response 1: We value your comment. We added a statement in our conclusion that the impact of telemedicine in the COVID-19 pandemic needs further research. The statement is the last paragrpah of the conclusion:

The COVID-19 pandemic quickly stimulated the use and expansion of telemedicine services in Latin America; however, further research is needed in order to determine the current effectiveness of the implementation of telemedicine in the region and recommend strategies for its continued expansion and regulation.

Point 2: Additionally, for a better manuscript, you can propose aspects in which inequalities is resolved by telemedicine.

Response 2: Thank you for your feedback; we value your suggestion. We took your advice and created a summary of relevant information, including aspects of inequalities resolved by telemedicine services by country. All this informaiton is shown in Table #1.

Point 3: You can provide damage report from COVID-19 in this region. From this damage, you must analyze needs of telemedicine compared with other regions. 

Response 3: Thank you for your input. While comparison of Latin America to other regions is out of the scope of this narrative review, we have incorporated in the last paragraph of section 3 “Health care quality in Latin America” some points about damage report in the region and how this is connected to the need for a better healthcare system. This paragraph can be read as following:

A fragmented healthcare system and varying rates of healthcare facilities and medical personnel make achieving healthcare for all in Latin American difficult. These shortcomings are reflected in the massive toll that the Latin American region paid with the COVID-19 pandemic. Latin America had 1.7 million deaths as of May 3, 2022 (over 27 percent of deaths worldwide). Deaths were highest in Brazil, Mexico, Peru, Colombia, and Argentina. To address unequal healthcare access, telemedicine offers a way to supplement coverage.

Point 4: Telemedicine-related laws over Latin America can be compared in a table for more readability. 

Response 4: Thank you for your input. To improve understanding, we have added Table #1 with a summary of relevant information, including telemedicine-related laws by country.

Point 5: In line 64, the reference style is different compared with other styles.

Response 5: Thank you for your comment. This has been addressed.

Point 6: Concrete examples of telemedicine usage can help the understanding of Latin America's status for other region's readers. In a current manuscript, writing style is somewhat strict. 

Response 6: We appreciate your comments, following this, we added examples of telemedicine usage by country in a brief summary. All this information is shown in Table #1.

Variable ratings.

Response: We have noted that you marked with 3/5 in the statement about the work being a significant contribution to the field. We expect this paper to be one starting point for other researchers who want to explore empirically telemedicine in Latin America. Information collected about telemdicine in Latin America thorughtout this work is mostly available only for spanish or Portuguese speakers.

We also noted that you marked with 4/5 in the statement about the work being well organized and about English being used correctly. Also, you rated the paper references adequacy and scientific soundess with 3/5. To improve readability and scientific soundeness, we hired a native English speaker academic copy editor. They extensively worked on the manuscript to improve its flow, readability, references relevance. and clearness.

Round 2

Reviewer 3 Report

This manuscript is improved sincerely in the aspect of easy reading and adding additional information by my comments. I recommend this paper's publication for this journal.  

Author Response

We thank Reviewer #3 for his comment. Their comments improved significantly our manuscript.